# Comprehensive analysis of male-free reproduction in *Monomorium triviale* (Formicidae: Myrmicinae)

**Naoto Idogawa**[1]*, **Tomonori Sasaki**[2,3], **Kazuki Tsuji**[3], **Shigeto Dobata**[1,4]*

**1** Laboratory of Insect Ecology, Graduate School of Agriculture, Kyoto University, Kyoto, Japan, **2** Research and Development Department, Fumakilla Limited, Hatsukaichi, Hiroshima, Japan, **3** Department of Subtropical Agro-Environmental Sciences, Faculty of Agriculture, University of the Ryukyus, Nishihara, Okinawa, Japan, **4** Department of General Systems Studies, Graduate School of Arts and Sciences, The University of Tokyo, Meguro-ku, Tokyo, Japan

* idogawa.naoto.na@alumni.tsukuba.ac.jp (NI); dobata@g.ecc.u-tokyo.ac.jp (SD)

**Data Availability Statement:** New COI sequences generated for this study are deposited in DNA Data Bank of Japan (DDBJ) under accession numbers LC592050 to LC592065. All the raw sequence data for microbial analysis have been deposited at the

## Abstract

We report comprehensive evidence for obligatory thelytokous parthenogenesis in an ant *Monomorium triviale*. This species is characterized by distinct queen–worker dimorphism with strict reproductive division of labor: queens produce both workers and new queens without mating, whereas workers are completely sterile. We collected 333 nests of this species from 14 localities and three laboratory-reared populations in Japan. All wild queens dissected had no sperm in their spermathecae. Laboratory observation confirmed that virgin queens produced workers without mating. Furthermore, microsatellite genotyping showed identical heterozygous genotypes between mothers and their respective daughters, suggesting an extremely low probability of sexual reproduction. Microbial analysis detected no bacterial genera that are known to induce thelytokous parthenogenesis in Hymenoptera. Finally, the lack of variation in partial sequences of mitochondrial DNA among individuals sampled from across Japan suggests recent rapid spread or selective sweep. *M. triviale* would be a promising model system of superorganism-like adaptation through comparative analysis with well-studied sexual congeners, including the pharaoh ant *M. pharaonis*.

## Introduction

Hymenopteran insects are characterized by haplo-diploid sex determination systems, in which females are derived from fertilized diploid eggs and males are derived from unfertilized haploid eggs via arrhenotokous parthenogenesis [1]. However, the production of diploid females from unfertilized eggs—known as thelytokous parthenogenesis—has been reported sporadically across the Hymenoptera [2]. Thelytoky has provided an opportunity for empirical testing of evolutionary hypotheses to account for the near-ubiquity of sex among eukaryotes [3]. The last decade has seen an increasing number of known thelytokous Hymenoptera, especially in eusocial species [4]. In studies taking advantage of this simplified mode of reproduction,

DDBJ SRA (DRA) under accession number DRA011730 (DRR279213-DRR279218).

**Funding:** This work was supported by a Japan Society for the Promotion of Science (JSPS) Research Fellowship for Young Scientists to NI (19J22242) and a grant from the Secom Science and Technology Foundation to SD. The funders had no role in study design, data collection and analysis, decision to publish, or preparation of the manuscript.

**Competing interests:** The authors have declared that no competing interests exist.

thelytokous social insects have been acknowledged as a model system in behavioral ecology (e.g., [5–8]) and sociogenomics (e.g., [9]).

Previous studies of thelytokous parthenogenesis of ants have identified three distinct categories [10, 11]: (type I) queens produce workers sexually but daughter queens via thelytoky; (type II) the queen caste is usually lost, males are absent, and workers produce workers via thelytoky; and (type III) queens produce both workers and queens via thelytoky, males are usually absent, and workers are sterile.

In this paper, we report comprehensive evidence of the third type of thelytokous parthenogenesis, namely obligatory thelytoky by queens of *M. triviale* [12]. The genus *Monomorium* is one of the most species-rich genera among ants [13] and has a worldwide distribution [14]. With the marked exception of tramp species such as the pharaoh ant *M. pharaonis* [15], most *Monomorium* species remain to be investigated [16]. *M. triviale* is reported from East Asia, encompassing Japan, South Korea and mainland China [14]. Neither males nor inseminated females have been found to date; this has been referred to in several studies [1, 4, 17–23]. Surprisingly, however, direct evidence of thelytoky is lacking. We examined the reproductive system of *M. triviale* through multiple approaches, namely dissection of spermathecae of field-collected queens, direct observation of virgin queen reproduction, and microsatellite genotyping of mothers and daughters. Furthermore, we analyzed for the presence of parthenogenesis-inducing microsymbionts and within-species phylogenetic relationships among populations in Japan.

## Materials & methods

### Colony sampling

A total of 333 nests of *M. triviale* were collected from 14 local populations in Japan (Fig 1, Table 1). No specific permission for sampling was required. In the field, nests were found in small gaps in dead plant bodies such as rotten roots, dead twigs, hollow bamboo sticks, and acorns. Because of the small nest size (about 200 workers on average) it was easy to collect the whole nest. During the field investigation, males of *M. triviale* were never found. Colonies were transferred into artificial nests in the laboratory as soon as possible and were kept at 25˚C until the following experiments. In addition, ethanol-preserved samples originating from three other sites were examined (Fig 1, nos. 15 to 17).

### Dissection of wild queens

To confirm the reproductive status of the queens from wild nests, we dissected 63 individuals from 38 nests of 10 *M. triviale* populations within 6 months after collection (Table 2). First the queen was immobilized by soaking in 70% ethanol for 3 min. The body was then transferred to a 30-mm petri dish filled with distilled water, and the internal reproductive organs were pulled out from the end of the abdomen by using precision forceps under a binocular microscope (SZ40; OLYMPUS Optical, Tokyo, Japan). The mating status of the queen was determined by the presence or absence of sperm in the spermatheca. As an indicator of oviposition, the ovary's yellow body was checked. As a positive control [18], 11 queens of a congeneric species, *M. intrudens*, were dissected in the same manner.

### Rearing experiment

Each wild nest was put into a plastic container (68 × 39 × 15 mm) with gypsum on the bottom. The nests were kept in the laboratory at 25˚C, and water and food were replenished every 3 days. The nests were fed mealworms, *Tenebrio molitor*, cut into approximately 5-mm lengths.

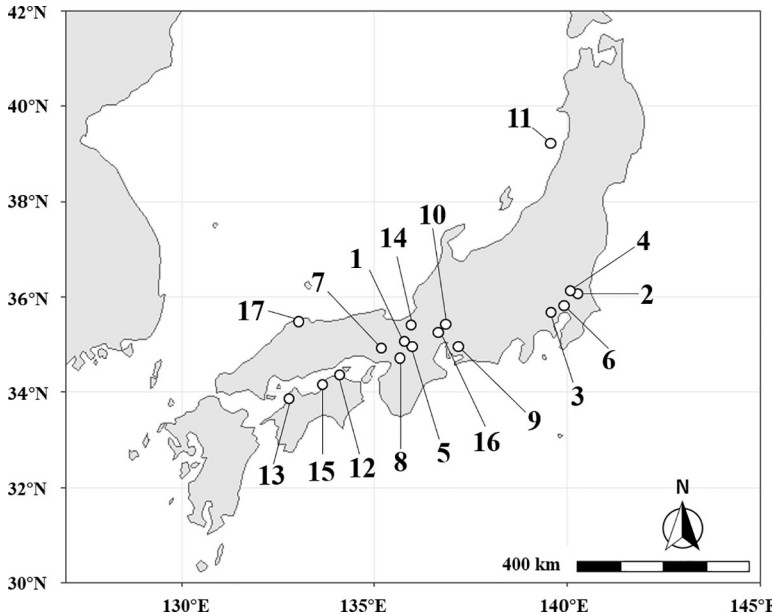

**Fig 1. Localization of the 14 sampling sites (1 to 14) and the three sites of origin of laboratory-reared populations (15 to 17) in Japan.** Open circles indicate site locations. Site names, numbers of nests collected, and types of analyses are described in Table 1. Map data by Natural Earth (http://www.naturalearthdata.com/).

**Table 1. Sampling sites of *Monomorium triviale* populations and experiments performed.**

| | Locality | Latitude | Longitude | Collection dates | No. of nests | Experiments |
|---|---|---|---|---|---|---|
| 1 | Kyoto City, Kyoto Prefecture | 35.060087 | 135.788488 | 19/04/2017 to 27/11/2017 (24 times) | 176 | SP, VR, SEQ |
| 2 | Tsuchiura City, Ibaraki Prefecture | 36.077927 | 140.165473 | 20/06/2017, 09/10/2018 | 28 | SP, VR, SEQ |
| 3 | Chofu City, Tokyo Metropolis | 35.668182 | 139.549061 | 16/06/2017, 27/09/2018 | 28 | SP, VR, SEQ |
| 4 | Tsukuba City, Ibaraki Prefecture | 36.100415 | 140.101297 | 19/06/2017, 08/10/2018 | 26 | SP, VR, SEQ |
| 5 | Otsu City, Shiga Prefecture | 34.970262 | 135.956026 | 10/06/2017 | 26 | SP, VR, SEQ |
| 6 | Matsudo City, Chiba Prefecture | 35.774787 | 139.899362 | 18/06/2017, 28/09/2018 | 22 | SP, VR, SEQ |
| 7 | Higashiosaka City, Osaka Prefecture | 34.665748 | 135.671267 | 09/11/2017 | 5 | SP, SEQ |
| 8 | Okazaki City, Aichi Prefecture | 34.941529 | 137.175594 | 03/09/2017 | 1 | SP, SEQ |
| 9 | Sanda City, Hyogo Prefecture | 34.914575 | 135.165764 | 09/05/2017 | 1 | VR, SEQ |
| 10 | Kaizu City, Gifu Prefecture | 35.22 | 136.63 | 06/06/2018 | 1 | SP |
| 11 | Tobishima Island, Sakata City, Yamagata Prefecture | 39.19 | 139.55 | 20/09/2018 | 2 | SP |
| 12 | Takamatsu City, Kagawa Prefecture | 34.364349 | 134.098205 | 17/06/2017 | 16 | SEQ |
| 13 | Matsuyama City, Ehime Prefecture | 33.845539 | 132.765722 | 19/09/2018 | 2 | SEQ |
| 14 | Takashima City, Shiga Prefecture | 35.3677 | 135.9168 | 30/05/2017 | 1 | SEQ |
| 15 | Kanonji City, Kagawa Prefecture | 34.12 | 133.66 | Reared in Laboratory | – | SEQ |
| 16 | Kagamihara City, Gifu Prefecture | 35.40 | 136.85 | Reared in Laboratory | – | SEQ |
| 17 | Matsue City, Shimane Prefecture | 35.47 | 133.05 | Reared in Laboratory | – | SEQ |
| | | | | **Total** | **333** | |

The nests from Kaizu City (locality no. 10) and Tobishima Island (no. 11) were provided by Dr. K. Ohkawara. Samples from Higashiosaka City (no. 7) and Takashima City (no. 14) were provided by Mr. K. Sadahiro and Dr. T. Nozaki, respectively. Ethanol-preserved samples from Kanonji City, Kagamihara City, and Matsue City (nos. 15 to 17) were provided by Dr. F. Ito. Experiments on the samples from the 17 sites are abbreviated as follows: SP: Dissection of wild queens' spermathecae; VR: Observation of virgin queen reproduction; SEQ: Phylogenetic analysis based on mtDNA sequencing.

**Table 2. Dissected queens from each sampling site.**

| | Locality | No. of nests | No. of queens | Insemination | | Yellow body | |
|---|---|---|---|---|---|---|---|
| | | | | yes | no | yes | unclear |
| 1 | Kyoto City, Kyoto Prefecture | 3 | 10 | 0 | 10 | 9 | 1 |
| 2 | Tsuchiura City, Ibaraki Prefecture | 8 | 8 | 0 | 8 | 7 | 1 |
| 3 | Chofu City, Tokyo Metropolis | 8 | 8 | 0 | 8 | 5 | 3 |
| 4 | Tsukuba City, Ibaraki Prefecture | 8 | 8 | 0 | 8 | 3 | 5 |
| 5 | Otsu City, Shiga Prefecture | 2 | 9 | 0 | 9 | 9 | 0 |
| 6 | Matsudo City, Chiba Prefecture | 4 | 8 | 0 | 8 | 8 | 0 |
| 7 | Higashiosaka City, Osaka Prefecture | 1 | 1 | 0 | 1 | 1 | 0 |
| 8 | Okazaki City, Aichi Prefecture | 1 | 8 | 0 | 8 | 8 | 0 |
| 10 | Kaizu City, Gifu Prefecture | 1 | 1 | 0 | 1 | 1 | 0 |
| 11 | Tobishima Island, Sakata City, Yamagata Prefecture | 2 | 2 | 0 | 2 | 1 | 1 |
| | **Total** | **38** | **63** | **0** | **63** | **52** | **11** |
| – | *Monomorium intrudens*, Kyoto City, Kyoto Prefecture | 4 | 11 | 10 | 1 | 11 | 0 |

The numbers of individuals possessing a yellow body and presumed to have experienced oviposition are also shown. As a positive control, data for the sexual congener *M. intrudens* are shown in the bottom row.

During July and August 2017, a total of 44 queen broods (larvae or pupae) from 21 nests of seven populations were produced (Table 3). Each queen was isolated with 10 nestmate workers in a plastic container ($36 \times 36 \times 14$ mm). These nests were kept under the same conditions as the source nests. Reproduction by these virgin queens was observed for 6 months. Finally, all remaining queens were dissected and confirmed to have no sperm in their spermathecae (Table 4).

## Microsatellite analysis

To provide genetic evidence of thelytoky, 33 queens (from 17 nests of seven populations; all used in the rearing experiment) were examined (Table 5). They were genotyped at microsatellite locus *Mp-1* (`f: GCCAATGGTTTAATCCCTCA; r: TCATACTGCGTGTGCCTTTC`), originally developed from *M. pharaonis* [24]. Daughter workers produced from these virgin queens (174 individuals in total) were also genotyped and were compared with their mothers. The thorax of each individual was crushed and placed in a 0.2-mL microtube filled with 100 μL of a DNA extraction reagent (PrepMan Ultra Reagent, Applied Biosystems, Foster City, CA, USA). The polymerase chain reaction (PCR) cocktail contained 1 μL of template DNA, 0.3 μL of 25 mM $MgCl_2$, 0.3 μL of 10 mM dNTPs, 1.5 μL of 10× PCR Buffer, 0.1 μL of 5 U/μL

**Table 3. Thelytokous worker production in the rearing experiment.**

| | Locality | No. of nests | No. of queens isolated | No. of queens surviving for 6 months | No. of workers produced |
|---|---|---|---|---|---|
| 1 | Kyoto City, Kyoto Prefecture | 6 | 10 | 6 | 17 |
| 2 | Tsuchiura City, Ibaraki Prefecture | 3 | 7 | 2 | 11 |
| 3 | Chofu City, Tokyo Metropolis | 3 | 5 | 5 | 26 |
| 4 | Tsukuba City, Ibaraki Prefecture | 3 | 5 | 5 | 28 |
| 5 | Otsu City, Shiga Prefecture | 3 | 8 | 8 | 52 |
| 6 | Matsudo City, Chiba Prefecture | 2 | 7 | 8 | 28 |
| 9 | Sanda City, Hyogo Prefecture | 1 | 2 | 2 | 15 |
| | **Total** | **21** | **44** | **36** | **177** |

**Table 4. Dissection of virgin queens from the rearing experiment.**

| | Locality | No. of queens dissected | Insemination | | Yellow body | |
|---|---|---|---|---|---|---|
| | | | yes | no | yes | unclear |
| 1 | Kyoto City, Kyoto Prefecture | 0[a] | NA | NA | NA | NA |
| 2 | Tsuchiura City, Ibaraki Prefecture | 2 | 0 | 2 | 2 | 0 |
| 3 | Chofu City, Tokyo Metropolis | 5 | 0 | 5 | 3 | 2 |
| 4 | Tsukuba City, Ibaraki Prefecture | 5 | 0 | 5 | 2 | 3 |
| 5 | Otsu City, Shiga Prefecture | 8 | 0 | 8 | 3 | 5 |
| 6 | Matsudo City, Chiba Prefecture | 7 | 0 | 7 | 2 | 5 |
| 9 | Sanda City, Hyogo Prefecture | 2 | 0 | 2 | 1 | 1 |
| | **Total** | **29** | **0** | **29** | **13** | **16** |

[a]We did not examine queens from Kyoto due to their death before dissection.

Taq DNA Polymerase (QIAGEN, Valencia, CA, USA), 0.2 μL of U19 fluorescent dye, and 1.0 μL of each primer pair, to which distilled water was added to make a total volume of 15.2 μL. The PCR program consisted of an initial step of 94˚C for 180 s, followed by 35 cycles of 94˚C for 30 s, 57˚C for 60 s and 72˚C for 60 s, with a final step of 72˚C for 10 min. PCR products were mixed with Hi-Di formamide and GS-600 LIZ size standard and were analyzed by using a 3500 Series Genetic Analyzer and GeneMapper 5.0 software (Applied Biosystems).

## Microbial analysis

To assess potential infection of *M. triviale* by thelytoky-inducing bacteria, we performed high-throughput amplicon sequencing by using whole bodies of adults. In August 2020, three *M. triviale* nests were collected in Takaragaike Park, Kyoto, Japan (lat 35.060087, long 135.788488). All colonies were transported to our laboratory and moved to plastic containers (68 × 39 × 30 mm) with gypsum on the bottom. They were maintained at 25˚C, and water replenishment and mealworm-feeding were performed every 3 days. In October 2020, ten queens and 100 workers were picked up from each nest. The queens or workers from each nest were stored as a group in acetone at –30˚C. The pooled individuals represented one biological replicate (i.e., three replicates per caste). Whole bodies in each replicate were air-dried and pooled in a 1.5-mL plastic tube, and DNA was extracted with a QIAamp DNA Micro Kit (QIAGEN). We followed the manufacturer's instructions and added extra steps of thermal cycling and lysozyme to the protocol to ensure the lysis of Gram-positive bacterial cell walls: After being

**Table 5. Microsatellite analysis data.**

| | Locality | No. of nests | No. of queens | No. of workers | Genotype | Prob. that all offspring were heterozygous |
|---|---|---|---|---|---|---|
| 1 | Kyoto City, Kyoto Prefecture | 4 | 4 | 15 | 222/234 | $(1/2)^{15} = 3.05 \times 10^{-5}$ |
| 2 | Tsuchiura City, Ibaraki Prefecture | 1 | 2 | 11 | 220/232 | $(1/2)^{11} = 4.88 \times 10^{-4}$ |
| 3 | Chofu City, Tokyo Metropolis | 3 | 5 | 26 | 222/232 | $(1/2)^{26} = 1.49 \times 10^{-8}$ |
| 4 | Tsukuba City, Ibaraki Prefecture | 3 | 5 | 28 | 220/232 | $(1/2)^{28} = 3.73 \times 10^{-9}$ |
| 5 | Otsu City, Shiga Prefecture | 3 | 8 | 52 | 220/228 | $(1/2)^{52} = 2.22 \times 10^{-16}$ |
| 6 | Matsudo City, Chiba Prefecture | 2 | 7 | 27 | 220/234 | $(1/2)^{27} = 7.45 \times 10^{-9}$ |
| 9 | Sanda City, Hyogo Prefecture | 1 | 2 | 15 | 222/238 | $(1/2)^{15} = 3.05 \times 10^{-5}$ |
| | **Total** | **17** | **33** | **174** | **–** | $\mathbf{(1/2)^{174} = 4.18 \times 10^{-53}}$ |

Genotypes at the *Mp-1* locus are shown. Under the assumption of sexual reproduction, the probability that all daughter workers would be heterozygous was calculated.

crushed in 180 μL ATL buffer, samples in plastic tubes first underwent two cycles of −80˚C for 30 min and 50˚C for 5 min; then, under room temperature, we added 2 μL of lysozyme from egg white (Nakalai Tesque, Kyoto, Japan; 20 μg/μL TE buffer) to each sample and incubated the samples at 37˚C for 30 min.

16S rRNA amplicon sequencing and the subsequent data analysis were performed according to in-house workflow by Bioengineering Lab. Co., Ltd. (Sagamihara, Kanagawa, Japan). The V4 region was amplified from 1 ng of template DNA by using ExTaq HS (Takara Bio, Otsu, Shiga, Japan) polymerase and the 515f–806r primer pair. Sequences were determined by using a MiSeq system with a MiSeq reagent kit v3 (Illumina, San Diego, CA, USA), which generated 2 × 300-bp paired-end reads.

Demultiplexing (on the basis of a perfect match with the primer sequences used), adapter trimming, and quality filtering (phred score ≥ 20; primer sequences, 50 bp on both 3′-ends, noise and chimeric reads were removed) of the paired-end reads were performed by using a Fastx toolkit (ver. 0.0.14, http://hannonlab.cshl.edu/fastx_toolkit/) and dada2 plugin for Qiime2 (ver. 2020.8) [25], resulting in representative sequences and operational taxonomic unit (OTU) tables. Assignment of appropriate taxa (confidence level > 0.7) was performed by using the default setting of the feature-classifier plugin for Qiime2 against the EzBioCloud 16S reference database (https://www.ezbiocloud.net/).

### Phylogenetic analysis

To investigate intraspecific diversity and determine the phylogenetic position of *M. triviale*, phylogenetic analysis was performed. A 639-bp region of the *cytochrome oxidase I* (*COI*) sequence was determined by using PCR with a combination of primers, namely LCO1490 (`GGTCAACAAATCATAAAGATATTGG`) and HCO2198 (`TAAACTTCAGGGTGACCAAAA AATCA`) [26]. One worker randomly chosen from each of 15 local populations (Table 1) of *M. triviale* and a single worker of the sexual species, *M. intrudens*, collected in Kihoku-cho, Mie, Japan (lat 34.21, long 136.33), were sequenced. Protocols for DNA extraction and the PCR mix were the same as for the microsatellite analysis. The thermal cycle consisted of an initial denaturation at 94˚C for 1 min, 35 cycles of denaturation at 94˚C for 30 s, annealing at 50˚C for 30 s, extension at 72˚C for 60 s, and a final extension at 72˚C for 10 min. PCR products were ethanol-precipitated and sequenced in both directions by using a BigDye Terminator v3.1 cycle sequencing kit on a 3500 Series Genetic Analyzer (Applied Biosystems). A total of 16 sequences were submitted to the DNA Data Bank of Japan under accession numbers LC592050 to LC592065; see Fig 3). To identify the phylogenetic position of *M. triviale* after recent advances in systematics of the subfamily Myrmicinae [27], a congeneric species, *M. pharaonis* (GenBank accession number GU710434) and two former *Monomorium* species, *Syllophopsis sechellensis* (EF609858) and *Erromyrma latinodis* (GU709833) were added to the analysis. The red imported fire ant, *Solenopsis invicta* (HQ928672), was used as an outgroup. The 16 sequences, together with those of the above four species, were aligned by using ClustalW [28] in MEGA 10.0 [29]. Maximum likelihood analyses were conducted as implemented in MEGA 10.0 by using a GTR+I+Γ model with node support assessed with 1000 bootstrap replicates.

## Results

### Dissection of wild queens

All 63 dissected queens of *M. triviale* had no sperm in their spermathecae (Fig 2, Table 2). Yellow bodies, suggestive of queen oviposition experience, were identified in 52 individuals across all 10 sampling sites. In contrast, 10 out of 11 *M. intrudens* queens dissected as positive

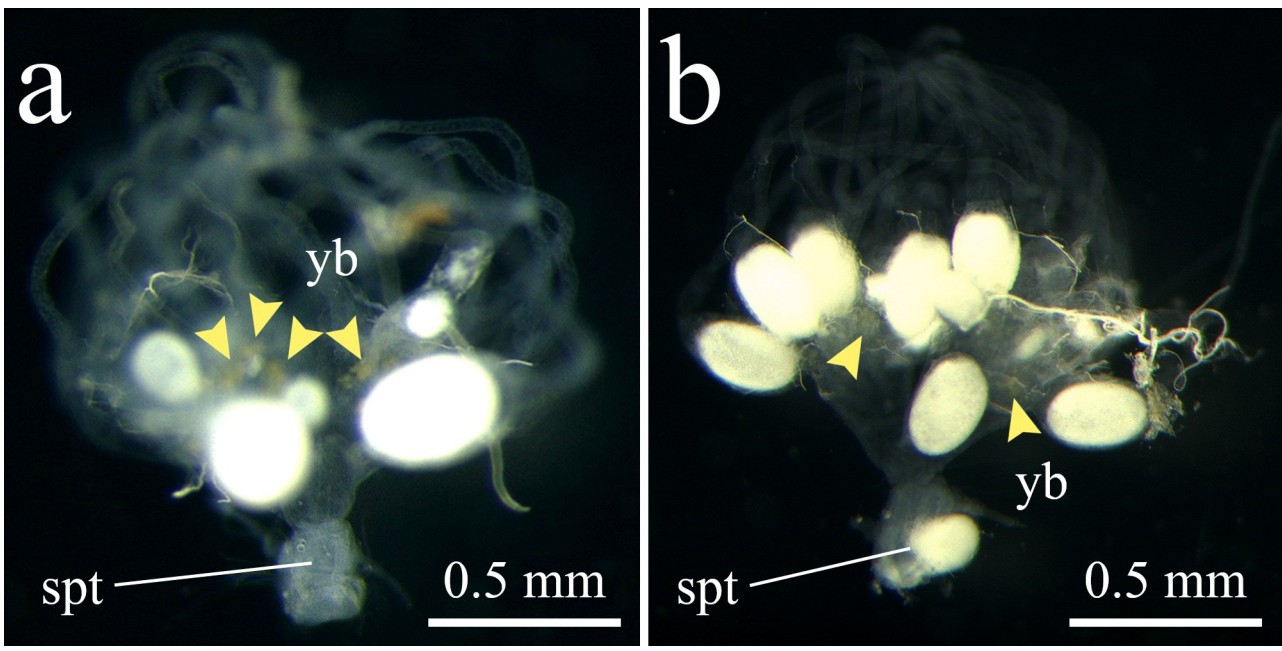

**Fig 2.** Reproductive organs of **(a)** *Monomorium triviale* and **(b)** *M. intrudens* queens. **(a)** *M. triviale*: Translucent (empty) spermatheca (spt), ovarioles with oocytes and obvious yellow bodies (yb) indicating that virgin queens were reproductively mature and laid eggs. **(b)** *M. intrudens*: Opaque (filled with sperm) spermatheca, well-developed ovarioles, and yellow bodies indicating that the queen is sexually reproducing. Yellow arrowheads indicate yellow bodies.

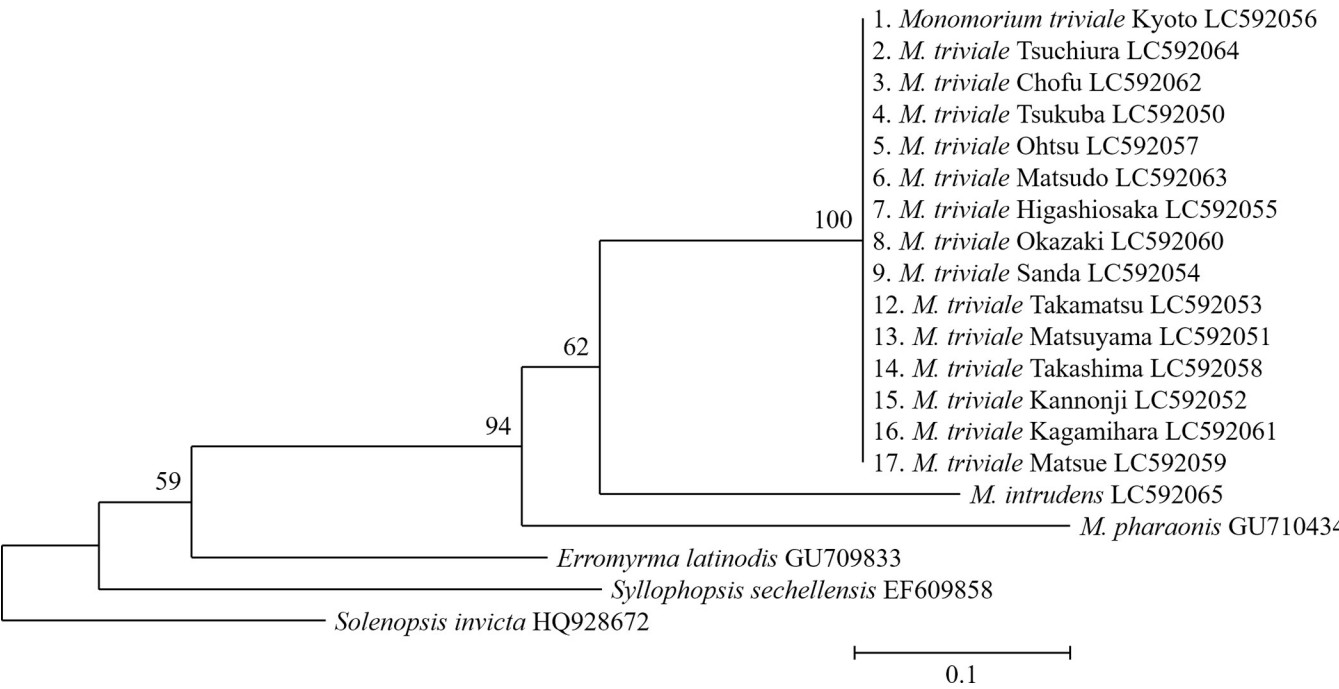

**Fig 3. Maximum likelihood tree of 15 populations of *M. triviale* and related species, based on *COI* sequences.** The number at each branch of the phylogenetic tree represents the bootstrap percentage (1000 replicates). GenBank accession codes follow the taxon names. Scale bar: 0.1 substitutions per site.

controls had sperm in their spermathecae. Dissection of *M. triviale* workers confirmed their complete sterility (S1 File), indicating that their complete sterility reported in previous studies [30, 31].

### Rearing experiment

Eight queens died before producing daughter workers (Table 3). Thirty-six virgin queens from seven local populations survived, and each produced at least one worker. In total, 177 workers emerged during the experimental period; no males were produced. All dissected queens had empty spermathecae (Table 4). Despite all the queens producing workers, only 13 of 29 queens dissected had obvious yellow bodies.

### Microsatellite analysis

Analysis of 33 queens and 174 workers showed that all individuals were heterozygous at *Mp-1* and that the genotypes of all the daughter workers were identical to those of their mothers. In addition, all individuals within the same local population had the same genotype. Comparison of genotypes between mothers and their daughters enables us to infer the occurrence of sexual reproduction. A potential male mate of a heterozygous mother (genotype AB) either shares or does not share the same allele (A or B) of *Mp-1* as the mother. We can rule out the latter possibility on the basis of the daughter genotypes. In the former case, the mother's sexually produced daughters should have two genotypes in the expected ratio of 1:1. One is the same as their mother's (AB) and the other is homozygous at one of the two mother alleles (AA when the father's genotype is A, or BB when the father's genotype is B). Under sexual reproduction, the observed bias toward the daughter's same heterozygous genotypes as their mother would be extremely rare. (The exact binomial probabilities are given in Table 5.) Therefore, we can reject the possibility of sexual reproduction among the individuals that we genotyped.

### Microbial analysis

MiSeq sequencing and Qiime2-based analysis yielded 8364 to 46,817 reads per biological replicate and a total of 267 OTUs (summarized in S1 Table). Among them, 263 OTUs (covering 99.9% to 100% of the reads) were classified as bacterial, and 117 OTUs (92.4% to 99.5% of the reads, excepting 54.7% of the reads from one queen replicate from nest B, see below) were classified at least to the phylum level. No reads were assigned to bacterial genera that are known to induce thelytokous parthenogenesis in Hymenoptera, i.e., *Cardinium*, *Rickettsia*, and *Wolbachia* [1].

In two nests (A and C) out of the three replicates for both queens and workers, the most abundant reads were classified as from *Spiroplasma platyhelix* (88.4% and 67.4% of total reads from queen and worker replicates, respectively, of nest A; 91.5% and 91.1%, respectively, for nest C). The replicates obtained from nest B showed a different pattern of *S. platyhelix* abundance: 0% from the queen replicate and 13.2% from the worker replicate. The absence of the *S. platyhelix* sequence from the nest B queen replicate was likely associated with the relatively small number of total reads (8364 vs. >30,000, S1 Table).

### Phylogenetic analysis

The 639-bp partial sequences of *COI* were completely identical among all individuals representing 15 populations of *M. triviale* (Fig 3). These individuals were placed in the monophyletic group with *M. intrudens* and *M. pharaonis*. Although the genus *Monomorium* has

recently been revealed to be a polyphyletic group [27, 32], our result suggests that *M. triviale* belongs to the genus *Monomorium sensu stricto*.

## Discussion

Our comprehensive investigation allows us to add *M. triviale* to the list of parthenogenetic ants. Production of daughters by unmated queens (indicative of type III thelytoky) was corroborated on the basis of multiple lines of evidence: (i) absence of males and inseminated queens in the field-collected nests; (ii) worker production by laboratory-reared virgin queens; (iii) unfilled spermathecae of queens that produced workers; and (iv) identical genotypes of mother queens and daughter workers. These features were common to all the locations we tested, suggesting that thelytoky is a dominant mode of reproduction across Japanese populations of *M. triviale*.

It should be noted here that rare occurrences of males have been reported in other parthenogenetic ants (type II), such as *Ooceraea biroi* [19] and *Pristomyrmex punctatus* [33]. In addition, queens of *M. triviale* retain undegenerated spermathecae [18], suggesting that they have low level of specialization to male-free reproduction. Moreover, geographic variation in sexual and asexual reproduction has been reported in some thelytokous ant species (types II and III), such as *Mycocepurus smithii*, *Myrmecina nipponica*, and *Platythyrea punctata* [17, 21, 34]. Whether and how often sexual reproduction occurs in *M. triviale*, especially in populations outside Japan, would be an interesting topic for future study.

Our exploratory analysis of bacterial communities in *M. triviale* provides basic information for future studies of the host–symbiont relationship. No evidence was found for infection with thelytoky-inducing bacteria in this species, confirming previous reports of no cases of microorganism-induced thelytoky in ants [10, 24, 35, 36]. *Spiroplasma* has been detected in various ant species [37], including another thelytokous species, *Mycocepurus smithii* (type III thelytoky) [38]. In *Solenopsis*, a genus closely related to *Monomorium*, *Spiroplasma* has been detected as a dominant bacterial taxon in whole bodies of adult workers of *S. geminata* [39], thus drawing parallels with our finding. It is noteworthy that type I thelytoky has recently been found in polygynous colonies of this species [40]. *Spiroplasma* is well known as a sex ratio distorter in *Drosophila* [41], and its role in the host's reproductive biology deserves further study in ants.

In our phylogenetic analysis, the extremely low levels of diversity observed in the *COI* sequences suggest the possibility of rapid spread or selective sweep in the recent past [42]. Having no mutation in 639 bp of COI sequence translates to an estimated divergence time of no more than ca. 45,000 years (assuming a divergence rate of 3.54% My$^{-1}$ [43]). Our results support the concept that this species is a member of the genus *Monomorium sensu stricto*. This phylogenetic status is advantageous in that the study designs established in *M. pharaonis*, a well-studied model system (e.g., [44–49]), will be applicable to future comparative analyses.

Thelytokous parthenogenesis is often considered as a trait overrepresented among introduced or invasive ant species [50]. Although some studies have categorized *M. triviale* as invasive [1, 19], no exotic distribution has been reported so far [14] and there is insufficient information to support the possibility of an introduced origin of the Japanese population of *M. triviale*. The known distribution range of this species is far more limited than those of successful invasive congeners such as *M. pharaonis* and *Monomorium floricola* [15, 51]. A preference for disturbed and urban habitats is shared widely among invasive ants [52]. Ito et al. [53] listed *M. triviale* along with *Strumigenys membranifera*, *P. punctatus*, and *O. biroi* as thelytokous ants found in "open, disturbed areas." Among these species, the life history of *M. triviale* is most poorly known. In our study, *M. triviale* nests were found even in a thicket dominated by

deciduous broad-leaved trees, which is not typical of "open, disturbed areas." Nevertheless, the above information does not rule out the possibility that *M. triviale* will become invasive in the future. Additional studies of this species' social structure, such as queen numbers and the mode of colony foundation, will help to evaluate the potential invasion risk of this species as an ecological consequence of its life history.

## Supporting information

**S1 File. Dissection of workers of *M. triviale*.**
(DOCX)

**S1 Table. Microbial taxa and their relative abundances (% of total reads) per each *M. triviale* sample.**
(XLSX)

## Acknowledgments

We are grateful to Kenji Matsuura who allowed us to use his laboratory. Fuminori Ito and Hroyuki Shimoji provided valuable advice on *M. triviale* biology and microbial analysis, respectively. We thank Kazuya Takeda, Kunio Sadahiro, Riou Mizuno and Yu Hisasue for field assistance. We also thank Hiroyuki Shimoji, Kiyomi Nakagawa, Kyosuke Ohkawara and Tomonari Nozaki for providing us with *M. triviale* nests.

## Author Contributions

**Conceptualization:** Naoto Idogawa, Kazuki Tsuji, Shigeto Dobata.

**Data curation:** Naoto Idogawa.

**Funding acquisition:** Naoto Idogawa, Shigeto Dobata.

**Investigation:** Naoto Idogawa, Tomonori Sasaki, Kazuki Tsuji, Shigeto Dobata.

**Methodology:** Naoto Idogawa.

**Resources:** Naoto Idogawa.

**Supervision:** Shigeto Dobata.

**Validation:** Tomonori Sasaki, Kazuki Tsuji.

**Visualization:** Naoto Idogawa.

**Writing – original draft:** Naoto Idogawa, Shigeto Dobata.

**Writing – review & editing:** Naoto Idogawa, Kazuki Tsuji, Shigeto Dobata.

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
