## [Decision Letter · Decision Letter 0]

15 Mar 2021

PONE-D-21-02452

Comprehensive analysis of male-free reproduction in Monomorium triviale (Formicidae: Myrmicinae)

PLOS ONE

Dear Dr. Idogawa,

Thank you for submitting your manuscript to PLOS ONE. After careful consideration, we feel that it has merit but does not fully meet PLOS ONE’s publication criteria as it currently stands. Therefore, we invite you to submit a revised version of the manuscript that addresses the points raised during the review process.

The two referees and I think your study is an important contribution to the ever growing field of alternative reproductive strategies in social insects. Only minor revisions are needed and I would be grateful if you were careful in taking them into account to make the paper even clearer and interesting for the reader.

We look forward to receiving your revised manuscript.

Kind regards,

Nicolas Chaline

Academic Editor

PLOS ONE

Journal Requirements:

In your Methods section, please provide additional information regarding the permits you obtained for the work. Please ensure you have included the full name of the authority that approved the sampling sites access and, if no permits were required, a brief statement explaining why.

3. We note that Figure 1 in your submission contain map images which may be copyrighted. All PLOS content is published under the Creative Commons Attribution License (CC BY 4.0), which means that the manuscript, images, and Supporting Information files will be freely available online, and any third party is permitted to access, download, copy, distribute, and use these materials in any way, even commercially, with proper attribution. For these reasons, we cannot publish previously copyrighted maps or satellite images created using proprietary data, such as Google software (Google Maps, Street View, and Earth). For more information, see our copyright guidelines: http://journals.plos.org/plosone/s/licenses-and-copyright.

3a, You may seek permission from the original copyright holder of Figure 1 to publish the content specifically under the CC BY 4.0 license. 

3b, If you are unable to obtain permission from the original copyright holder to publish these figures under the CC BY 4.0 license or if the copyright holder’s requirements are incompatible with the CC BY 4.0 license, please either i) remove the figure or ii) supply a replacement figure that complies with the CC BY 4.0 license. Please check copyright information on all replacement figures and update the figure caption with source information. If applicable, please specify in the figure caption text when a figure is similar but not identical to the original image and is therefore for illustrative purposes only.

Reviewers' comments:

Reviewer's Responses to Questions

**Comments to the Author**

1. Is the manuscript technically sound, and do the data support the conclusions?

Reviewer #1: Yes

Reviewer #2: Yes

2. Has the statistical analysis been performed appropriately and rigorously? 

Reviewer #1: Yes

Reviewer #2: Yes

3. Have the authors made all data underlying the findings in their manuscript fully available?

Reviewer #1: Yes

Reviewer #2: Yes

4. Is the manuscript presented in an intelligible fashion and written in standard English?

Reviewer #1: Yes

Reviewer #2: Yes

5. Review Comments to the Author

Reviewer #1: Idogawa et al. have done a very nice job documenting obligate thelotoky in Monomorium triviale. The work is very thorough using multiple approaches that cover all the bases of what one would want to see in a study of this nature. Analyses of additional microsatellite markers would have made it a little stronger, but the results are very solid as is. I appreciated the phylogenetic analysis given the recent upheaval regarding Monomorium taxonomy, as well as the microbial analysis to rule out a bacterial influence underlying asexual reproduction in this species. This study allows us to add another species to the list of obligate thelotokous ants, most specifically, those with queens that produce both new queens and workers parthenogenetically. The different approaches and the results are clear and easy to follow. It is very well written and I’m confident that it will be of general interest to readers of PLoS ONE. I have only very minor suggestions to help improve the paper.

Line 95: consider modifying to “First the queen was immobilized by soaking in 70% ethanol for 3 min.”

Line 254: should ref. 26 be ref. 27?

Line 282: consider rewording to “derived from the same daughter cell during the first meiotic division are fused to restore diploidy after the second meiotic division.”

Similarly line 284 could be reworded to read “different daughter cells derived during the first meoitic division are fused.”

Line 286: should read “hymenopteran” as it is used as an adjective here rather than a noun.

Line 312: consider replacing “habitat” with “habitats”

Line 318: consider replacing “Future” with “Additional” to avoid repetition since the last word of the previous sentence is “future.”

Excellent paper!

Reviewer #2: The occurrence of clonal reproduction in the ant Monomorium triviale has been regularly mentioned in previous publications, but no supporting data has yet been published. In this manuscript, Idogawa and coauthors demonstrate that Japanese populations of this species reproduce by clonal reproduction using several lines of evidence (dissection of queens' spermatheca, observation of laying by virgin queens, and microsatellite analysis) and that no known parthenogenesis-inducing bacteria could be found in colonies.

This is a usefull contribution to the field. The methods are sound and the results mostly support the authors' conclusion. One may regret, however, that no data demonstrating worker sterility is shown or cited (i.e. dissection of workers or rearing of queenless colonies). Besides, the discussion regarding the mode of parthenogenesis employed by M. triviale should be reconsidered. The authors suggest that clonal reproduction may proceed through central fusion automixis based on the observation that heterozygosity at the studied microsatellite is maintained across generations. Yet, this pattern could also result from apomictic parthenogenesis or terminal fusion automixis with selection against homozygosity.

Minor comments:

- There are some inconsistencies in Table 1 (there are more surviving queens than isolated queens in locality 6) and Table 4 (no locality 1 is not mentioned)

- l76-77 and l86-88: That "ethanol-preserved samples" are only mentioned for colonies maintained alive sounds weird.

- l300-302: One can be more precise here. Having no mutation in 639 bp of COI sequence translates to an estimated divergence time of no more than 45,000 years (assuming a divergence rate of 3.54% My−1; https://doi.org/10.1093/molbev/msq051).

Hugo Darras and Laurent Keller

6. PLOS authors have the option to publish the peer review history of their article (what does this mean?). If published, this will include your full peer review and any attached files.

Reviewer #1: No

Reviewer #2: **Yes: **Hugo Darras and Laurent Keller

---

## [Author Response · Author response to Decision Letter 0]

24 Mar 2021

Dear Dr. Chaline,

We are very pleased to have your valuable comments on the manuscript entitled “Comprehensive analysis of male-free reproduction in Monomorium triviale (Formicidae: Myrmicinae)" (PONE-D-21-02452). Based on the important advice from the two reviewers, we have made corrections as much as possible, and the following revisions have been made. Newly added supporting information S1 File responded to comments from reviewer #2 by confirming complete worker sterility in response to. We hope this revised manuscript could now be acceptable for publication. 

Sincerely,

Naoto Idogawa and Shigeto Dobata

Response to Reviewers

We appreciate your constructive advice and have incorporated them into our manuscript as much as possible, as described below. In the file derived from the original WORD file, the revised sentences are highlighted, including additional acknowledgement. Lines in your comments indicate those of the old version you have already read, and those in the revised manuscript are indicated by line numbers in the WORD file.

Journal Requirements

[J-1] Please ensure that your manuscript meets PLOS ONE's style requirements, including those for file naming.

I&D: We ensured that our manuscript including file naming meets PLOS ONE's style requirements.

[J-2] Please ensure you have included the full name of the authority that approved the sampling sites access and, if no permits were required, a brief statement explaining why.

I&D: We added a brief statement in the Materials & Methods (Lines 71) that no specific permission for sampling was required.

[J-3] We note that Figure 1 in your submission contain map images which may be copyrighted. (~~) We require you to either (1) present written permission from the copyright holder to publish these figures specifically under the CC BY 4.0 license, or (2) remove the figures from your submission.

I&D: In our Figure 1, we used map data from Natural Earth (public domain: http://www.naturalearthdata.com/). We indicate the data source credit in figure legend (line 81-82).

[J-4] Please review your reference list to ensure that it is complete and correct.

I&D: We checked the reference list as much as possible.

Reviewer #1

We highly appreciate your valuable comments. We made the following revision:

[1-1] Line 95: consider modifying to “First the queen was immobilized by soaking in 70% ethanol for 3 min.”

I&D: We modified the sentence accordingly (line 95).

[1-2] Line 254: should ref. 26 be ref. 27?

I&D: Thank you for your thorough reading. The citation number has been corrected (line 256).

[1-3] Line 282: consider rewording to “derived from the same daughter cell during the first meiotic division are fused to restore diploidy after the second meiotic division.”

[1-4] Similarly line 284 could be reworded to read “different daughter cells derived during the first meiotic division are fused.”

[1-5] Line 286: should read “hymenopteran” as it is used as an adjective here rather than a noun.

I&D: We appreciate your suggestion. Following comments from reviewer #2, we removed this paragraph.

[1-6] Line 312: consider replacing “habitat” with “habitats”

I&D: We modified the sentence accordingly (line 307).

[1-7] Line 318: consider replacing “Future” with “Additional” to avoid repetition since the last word of the previous sentence is “future.”

I&D: We modified the sentence accordingly (line 313).

Reviewer #2

We highly appreciate your constructive suggestions. We made the following revision:

[2-1] One may regret, however, that no data demonstrating worker sterility is shown or cited (i.e. dissection of workers or rearing of queenless colonies).

I&D: We added the data supporting obligate worker sterility by dissection as a supporting information. 

[2-2] Besides, the discussion regarding the mode of parthenogenesis employed by M. triviale should be reconsidered. The authors suggest that clonal reproduction may proceed through central fusion automixis based on the observation that heterozygosity at the studied microsatellite is maintained across generations. Yet, this pattern could also result from apomictic parthenogenesis or terminal fusion automixis with selection against homozygosity.

I&D: We agree with your comment, our discussion about the mode of parthenogenesis is too speculative. Therefore, we removed this paragraph.

[2-3] There are some inconsistencies in Table 1 (there are more surviving queens than isolated queens in locality 6) and Table 4 (no locality 1 is not mentioned)

I&D: Thank you for your thorough reading. We understand that you mentioned Table 3 and 4 and corrected them.

[2-4] l76-77 and l86-88: That "ethanol-preserved samples" are only mentioned for colonies maintained alive sounds weird.

I&D: We agree with your comment. We modified the sentences (line 76-77 and 87-88).

[2-5] l300-302: One can be more precise here. Having no mutation in 639 bp of COI sequence translates to an estimated divergence time of no more than 45,000 years (assuming a divergence rate of 3.54% My−1; https://doi.org/10.1093/molbev/msq051).

I&D: We appreciate your suggestion: 1,000,000 * (1/639)/0.0354 ~ 45,000. We added (line 295-296) together with the relevant reference.

---

## [Editor Report · Decision Letter 1]

29 Mar 2021

Comprehensive analysis of male-free reproduction in Monomorium triviale (Formicidae: Myrmicinae)

PONE-D-21-02452R1

Dear Dr. Idogawa,

We’re pleased to inform you that your manuscript has been judged scientifically suitable for publication and will be formally accepted for publication once it meets all outstanding technical requirements.

Kind regards,

Nicolas Chaline

Academic Editor

PLOS ONE
---

## [Editor Report · Acceptance letter]

13 Apr 2021

PONE-D-21-02452R1 

Comprehensive analysis of male-free reproduction in *Monomorium triviale* (Formicidae: Myrmicinae) 

Dear Dr. Idogawa:

I'm pleased to inform you that your manuscript has been deemed suitable for publication in PLOS ONE. Congratulations! Your manuscript is now with our production department. 

Kind regards, 

on behalf of

Professor Nicolas Chaline 

Academic Editor

PLOS ONE